

# 1 ESD Ideas

# 2 It is not an Anthropocene; it is really the
# 3 Technocene: names matter in decision
# 4 making under Planetary Crisis.

Oliver López-Corona[1,2,*] and Gustavo Magallanes-Guijón[3,4,*]
*¹Cátedras CONACyT, Comisión Nacional para el Conocimiento y Uso de la Biodiversidad (CONABIO), CDMX,*
*México*
*²Centro de Ciencias de la Complejidad (C3), Universidad Nacional Autónoma de México, CDMX, México.*
³Posgrado en Astrofísica, Instituto de Astronomía, *Universidad Nacional Autónoma de México, CDMX, México.*
⁴*Facultad de Ciencias, Universidad Nacional Autónoma de México, CDMX, México.*
*lopezoliverx@otrasenda.org
*gustavo.magallanes.guijon@ciencias.unam.mx

## 16 Abstract

We do not understand what we see but see what we understand. Words shape the comprehension of our
environment and set the space of possibilities we can access when decision making. Inhere we make the
case for the use of Technocene instead of Anthropocene using well-grounded arguments in basic
scientific principles. We already know that the Earth system has co-evolved with life phenomena (i.e. the
evolution of atmosphere chemistry). What the Technocene idea makes clear is that as modern human
societies exhibit an enormous coupling with technology and for the first time in human history that
technology has the potential to modify the very core processes that drive Earth System dynamics, then
Technology most be considered as a new dimension of analysis in the study of Earth system in its co-
evolution with life and particularly human beings.
27  ---
Earth is a complex system, that is maintained in a unique state far from thermodynamic equilibrium
through the co-evolution of its biotic and abiotic components by maximizing the entropy production, a
process that might be a thermodynamic imperative (Kleidon, 2009; Michaelian, 2012). The Evolution by
natural selection consider one direction of this coupling but the other direction, Niche Construction, has
been little studied. In previous work (López-Corona et al, 2019) we developed a new ontology, the
Ecobiont, to take both directions into account.
The theoretical model for the Ecobiont ontology considers a set of interacting pools: genes(G),
microbiome (g) and social (s); that co-evolve from some arbitrary time t to t', through natural selection




and niche construction. In contrast of how an abiotic component of Earth system evolve from a pool of
physicochemical components, biological and human systems do it with information stored not only in the
genome (physiochemical component) but also in its culture, including technology.
Considering this, we porpoise the following co-evolutionary multidimensional process
$$[(G1\ X\ g1\ X\ s1)\ X\ E1]\ X...\ X\ [(Gn\ X\ gn\ X\ sn)\ X\ En]\ \text{-->}\ [(G'1\ X\ g'1\ X\ s'1)\ X\ E'1]\ X...\ X\ [(G'n\ X'\ g'n\ X\ s'n)\ X\ E'n] \quad (1)$$
where $G_i$ is the genotype of the population i that is coupled with its symbionts ($g_i$) and together as an
Holobiont, including the social pool, co-evolve with local environment $E_i$ forming one coherent
evolutionary unit; which in turn co-evolve in parallel with many other of this units or Ecobionts.
To make it clear, it is no longer only a matter of genome or even culture, now it is mainly a
matter of how technology modify the evolutionary processes and even Earth System directly (i.e.
Climate Change or Ozone Layer Depletion). Then, in order to fully understand the current
planetary crisis and make good decisions about how to respond to it, we must be aware of this
new extra and key dimension. In our framework, this will lead to a special kind of Ecobiont that
capture the existence of human societies extreme coupled with technology.
Considering that Burger and co-workers (2017) have shown that *Homo Sapiens* living in modern
cities fall out extra-metabolic energy scaling every other mammal do-follow, including hunter-
gatherers that we called *Classical Homo Sapiens* (CHS), we proposed that those *Homo*
*Sapiens* living in modern cities are in fact a different type of Ecobiont compared with CHS, we
called them: Technobionts. This new (in geological time) Ecobiont type has turned itself into a
dynamic driver for earth functioning that has overwhelming the great forces of nature (Steffen et
al, 2007).
Because of the above, here we propose that the Anthropocene new geological era, that is about to get
formal recognition, is not the concept we need. For thousands of years CHS co-existed with the rest of the
Earth system's components (biotic and abiotic), so the ongoing Climate or Biodiversity (Dirzo, et al,
2014) Crisis are not caused by our human (Anthropic) nature but by an over coupling with some kinds of
technologies that enhance unprecedented niche construction capacities.
"What's in a name? That which we call a rose by any other name would smell as sweet." This phrase—
from William Shakespeare— is one of the clearest examples of the role that labeling exerts to shape
human perception. That's why selecting the name for this new era is key.
Technocene responds to the correct source of our current planetary crisis and point out to the proper path,
not stop being humans or accepting the catastrophe as Anthropocene would imply, but to find
configurations of technologies that take us back to the CHS track as possible, and away from tipping
points that could transform the Earth System in irreversible way (Steffen et al, 2018).
For example, in terms of Anthropocene, a solution to Planetary Crisis could be preferably searched into
technologies such as Geoengineering which is regarded by advocates as a creative and responsible
technological option in the face of a Climate Crisis (Thiele, 2019). Nevertheless, these calls for



emergency geoengineering need to be analyzed with extreme care in a full interdisciplinary or even transdisciplinary manner (Blackstock and Low, 2018) because this kind of re-coupling with new unproven technologies could carry out hidden systemic risk, so Precautionary Principle should prevail (Taleb et al, 2014).

On the other hand, a Technocene perspective could certainly promote technology de-coupling or at least a higher level of technology selection, promoting less invasive ones. For example, in terms of Climate Crisis society could embrace voluntary resignation to certain types of energy use to match sustainable energy budgets like the one promoted by McKay (2008).

Consider the profound impact to Earth System dynamics that came from the emergence of the 3,7000 mile planetary scare we know as the East African Rift Valley some eons ago, or how about some 4 million years ago, grasslands began to replace thick forests, and a dramatic pattern emerged in which  our ancestors adapted to the unstable environment by increasingly inventive use of technology and enhanced social cooperation (Dartnell, 2019). So, should we be concern about, for example, the results by Wang and Su (2019) who has showed a suggesting chain of evidence that both ML5.7 and ML5.3 earthquakes from 2018 in Sichuan Province China were induced by nearby Hydraulic Fracking activities?

 In that sense, potential awareness induced by recognizing over technological coupling in Technocene or Technobiont concepts could lead to a more precautionary use of some technologies. The Technocene concept is well-grounded into evolutionary and Earth System Dynamics theories, pose a better set mind for decision making and bottom line, we sure cannot stop being *Anthropos* but we may certainly stop being Technobionts.

In addition, thinking of Technocene rather than Anthropocene, also opens debate and analysis of philosophical (ontological, ethical), political and social problems about Climate Change and other components of Planetary Crisis, enhancing a deeper integral understanding of it.

Finally, beyond this conclusion around Planetary Crisis and decision making, we consider that Technocene framework highlight the co-evolutionary processes driven by natural selection and niche construction with a remarkable effect of some technologies. It also points to some very interesting theoretical possibilities because bottom line, it might be interpreted as a contextual statistical perspective of Earth System dynamics. Statistical contextually was developed mainly by Khrennikov (2009) as a modification of classical Kolmogorovian probability, that work as a formal framework for systems that are so context dependent that they should no longer be represented separated from it and then a new basic unity of analysis should be the indivisible pair (system, context).  In this sense, what we would be suggesting is that because the potential planetary impacts modern human societies (over coupled with some technologies) have, any Earth System dynamics description is incomplete without the human technological context.

# Acknowledgment

This work was supported by CONACyT fund M0037-2018-07, number 296842; Cátedras CONACyT fellowship program (project number 30); Sistema Nacional de Investigadores SNI, numbers 62929.



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
