# Peer review of "It is not an Anthropocene; it is really the"

_Earth System Dynamics, 2019_

## Short Comment (SC1) · 29 Dec 2019

Dear Editor,

The authors make an original and valuable elaboration about the term technocene, providing a very short rationale and examples for its necessity. I believe the authors just outline some ideas here, and the material is worth exploring in a larger and more complete piece. Personally I do not like the term Technocene because there is a trend of creating a profusion of fancy terms that might unnecessarily shift the focus towards

an unproductive side, however the authors make some fundamental points that are worth of attention.

Indeed, it is the last 50 years of accelerated technology use that created the so-called great acceleration. Technology (broadly defined) and disruption are going hand in hand across scales in the Earth system, and there is not a sufficient focus on this topic, hence I recommend publishing this manuscript after the authors perform a major revision as suggested below.

Suggestions about the paper:

I do not see the need of the equations and the emphasis on the genes, but the authors can perhaps improve the added value of these elements in the paper.

"we called them: Technobionts" do the authors refer to an existing publication of them clarifying these terms? If so please provide the details of the reference.

"in terms of Anthropocene, a solution to Planetary Crisis could be preferably searched into technologies such as Geoengineering". Preferably by whom? Is this about the preference of humans for either distributed renewables or geoengineering? This seems a weak point.

The elaboration about the Rift and Sichuan should be aligned better with the whole argument, the substance seems right for the argument, but it is still unrelated.

The last paragraph needs a grammar check and perhaps re-writing into clearer sentences.

The topic requires a larger and more complete piece with more organised arguments, flow and examples.

---

## Author Comment (AC1) · 31 Dec 2019

Dear Prof. Roger Cremades, we are greatly thankful to you for your valuable insightful commentaries that we have taken into account and have responded below.

Comments (»)

»I do not see the need of the equations and the emphasis on the genes, but the authors can perhaps improve the added value of these elements in the paper.

[Figure]

It was also the opinion of the editor. We agree and have removed it

»"we called them: Technobionts" do the authors refer to an existing publication of them clarifying these terms? If so please provide the details of the reference.

That is in fact the case: https://www.researchers.one/article/2019-01-1 which is within the reference but was not adequately cited when call technobots in the sentences of the comment, we have cited accordingly now.

»"in terms of Anthropocene, a solution to Planetary Crisis could be preferably searched into technologies such as Geoengineering". Preferably by whom?

We consider it would be preferably by decision makers that are prone to technological solutions.

»Is this about the preference of humans for either distributed renewables or geoengineering?

No, we tried to make the point that by using the concept of Anthropocene, we may be hiding the importance of technology as a source of the crisis, so focusing too much into technological solutions may get us into a never ending circle of problems made by abuse of technology that are tried to be fixed by using more technology that would lead to new problems (maybe even worst problems). We use geoengineering just as an example, we think a good example of technology with systemic (planetary) risks.

»This seems a weak point.

We re-written the paragraph to make this more clear: For example, in terms of Anthropocene that does not explicitly acknowledge the current key role of technology but only its human origin, a solution to Planetary Crisis may be searched into the technology itself in some sort of red queen process, as not identified as an important component of the problem. This would be similar to trying to resolve antibiotic bacteria resistance problems only by looking for better antibiotics (technological focus) without understanding that abuse in the use of antibiotics (technology) is a big part of the problem. Focusing too much on technological solutions may get us into a never-ending circle of problems made by abuse of technology that is meant to be fixed by using more technology that would lead to new problems (maybe even worst problems). In particular, there has been recent attention to Big Solutions approach in terms of for example geoengineering, which is regarded by advocates as a creative and responsible technological option in the face of a Climate Crisis (Thiele, 2019). Nevertheless, these calls for emergency geoengineering need to be analyzed with extreme care in a full interdisciplinary or even transdisciplinary manner (Blackstock and Low, 2018) because this kind of re-coupling with new unproven technologies could carry out hidden systemic risk, so Precautionary Principle should prevail (Taleb et al, 2014).

»The elaboration about the Rift and Sichuan should be aligned better with the whole argument, the substance seems right for the argument, but it is still unrelated.

We agree, as it is a ESD-ideas paper we tried to be very short, but it might be too short. We have developed the point as follows: Planetary changes have occurred several times on Earth System, modeling not only its dynamics but also life evolution. Consider the profound impact to Earth System dynamics that came from the emergence of the 3,700-mile planetary scare we know as the East African Rift Valley some eons ago, or how about some 4 million years ago, grasslands began to replace thick forests, and a dramatic pattern emerged in which our ancestors adapted to the unstable environment by increasingly inventive use of technology and enhanced social cooperation (Dartnell, 2019). Because normally these changes take very long periods, we tend to ignore them from the human perspective, but when talking about planetary-scale technologies these changes could take only a few years. So, should we be concerned about, for example, the results by Wang and Su (2019) who has showed a suggesting chain of evidence that both ML5.7 and ML5.3 earthquakes from 2018 in Sichuan Province China were induced by nearby Hydraulic Fracking activities? Again, planetary scale technologies should always be considered under the Precautionary Principle.

»The last paragraph needs a grammar check and perhaps re-writing into clearer sentences.

Thank you for pointing this out, we changed the paragraph as follows: Finally, beyond this conclusion around Planetary Crisis and decision making, we consider that Technocene framework highlights the co-evolutionary processes driven by natural selection and niche construction, turning attention to a topic that has not received enough consideration, the great technological acceleration of the past 50 years and how it has become an Earth system dynamics changer.

It also points to some very interesting theoretical possibilities because bottom line, it might be interpreted as a contextual statistical perspective of Earth System dynamics. Statistical contextually was developed mainly by Khrennikov (2009) as a modification of classical Kolmogorovian probability, that works as a formal framework for systems that are so context-dependent (coupled) that they should not be addressed separated but by an indivisible pair (system, context). In this sense, what we are suggesting is that because the potential planetary impacts modern human societies (over coupled with some technologies) have, any Earth System dynamics description is incomplete without the human technological context.

»The topic requires a larger and more complete piece with more organised arguments, flow and examples.

Maybe the references provided and the modifications be sufficient to keep this paper in the ESD-Ideas format which is very short.

---

## Referee Comment (RC1) · Axel Kleidon (Referee) · 28 Jan 2020

I am posting the following on behalf of an anonymous reviewer.

Axel Kleidon, Editor

====

This Ideas piece for ESD argues that the Anthropocene should in fact be called the Technocene. They suggest that the driver of the Earth-system changes that go under

the name of the Anthropocene is not in fact the Classical Homo Sapiens (CHS) of the Pleistocene and Holocene but is 'Homo Sapiens living in modern cities', which they say are a different type of 'ecobiont' (any coevolving set of genome, symbionts and milieu) that they call the 'Technobiont', to indicate that technology is a novel integral part of this ecobiont. They suggest that such a nomenclature will give a clearer signal that growing ecological problems 'are not caused by our human (Anthropic) nature but by an over coupling with some kinds of technologies that enhance unprecedented niche construction capacities.'

I have a lot of sympathy with the idea that technology is so central to the phenomenon of the Anthropocene that it should feature in the name that we give it. And 'the Technocene' is not a bad name, and has been used in a number of publications, for example by the anthropologist Alf Hornborg and the sociologist Hermínio Martins. I also think that there are some interesting ideas sketched here about how to theorise humans and their technology as a single evolving assemblage, using natural selection and niche construction.

However this piece would need a lot of revision before publication. Firstly, the details of the argument are too sketchy. The authors miss the chance to reference and summarise other literatures that support the idea that the relevant entity that might be pushing the Earth into a new system state is an assemblage comprising biological humans plus the exteriorisations of culture and technology. Some better examples would help too – that of fracking and Earth tremors doesn't seem to work so well, as it is not evidence of a proper systemic shift; there are many better examples in the canonical Anthropocene science papers, such as the shaping of rivers (e.g. http://dx.doi.org/10.1016/j.ancene.2015.03.003).

Secondly their normative conclusion – that we therefore need to reject technology, revert to classical Homo Sapiens, and thereby stay in the Holocene – does not follow. On the contrary, their argument (that technologically enhanced humans are simply the latest example of a long pattern of evolutionary shifts involving natural selection

and niche construction that can be expressed in a single formula (line 44)) seems to naturalise the current situation, assimilating it in a wider pattern in Earth history. To conclude that we need to reject technology and stay in the Holocene seems out of step with this idea – rather like imagining that when multicellular eukaryotes evolved, it would obviously have been better if they had all reverted back into being monocellular prokaryotes. I'm sure there are good reasons to be cautious about adopting certain new technologies such as solar radiation management climate geoengineering, but I remain unconvinced that the holobiont argument alone, or the Technocene name, are that helpful for identifying and assessing them. Indeed, one could argue that calling the new multi-thousand year geological epoch the Technocene rather than the Anthropocene will do more harm than good, by seeming to cement the idea of the rule of technology for millennia to come!

On more minor matters, the English language needs work. Most sentences have at least something wrong, but the worst slips are 'Inhere' instead of 'in here' (line 17 – though in fact it would be better to delete the word altogether), 'fall out' instead of 'follow' (line 58), 'scare' instead of 'scar' (line 94).

The Shakespeare quote, 'What's in a name? That which we call a rose by any other name would smell as sweet' (line 71) seems very ill-chosen by the way – the quote is generally taken to mean that names are not important, which is in tension with the basic idea of the article (lines 17-18) that names shape perception and cognition, so really matter.

---

## Author Comment (AC2) · 29 Jan 2020

We thank the anonymous reviewer for her/his comments that we respond below.

Comments (»>)

»> And 'the Technocene' is not a bad name, and has been used in a number of publications, for example by the anthropologist Alf Hornborg and the sociologist Hermínio Martins.
The reviewer is right we don't want to extent to much de article since it is in ESD-idea format (very limited number of words) but we included a pair of paragraphs citing them and others which appears in a quick scholar search using "technocene" in the title: https://scholar.google.com.mx/scholar?as_sdt=1,5&q=allintitle:+technocene&hl=en&as_vis=1

New text included:

"The concept of thechnocene is now new as it has been used by anthropologist Alf Hornborg (Hamilton, 2015) and the sociologist Hermínio Martins (2018) whose make a historical and critical review of the role of science in technological development, reviewing the relationship between nature and society from an interdisciplinary perspective. His analysis, which examines the need for political ecology, environmental anthropology and the relationship between science and society, is valuable for understanding different concepts such as the one proposed in this letter from the different disciplines: the Technocene.

In this ESD-Idea paper, we use the notion of Technocene as an environmental concept, in which Environmental Sciences, Earth Sciences converge, and as Hamilton (2015) says the post-Cartesian Social Science, too. At the same time that the interdiscipline of science is intertwined to explain environmental phenomena such as global warming.

For us it is clear that thinking about the concept of Anthropocene is exceeded in the face of technological development and its environmental impact (Cera, 2017). So treating environmental problems and their research from an anthropocentric approach is not adequate.

On the other hand, ethical and political problems must be treated in their right dimension (Hensel, 2017), and for this it is necessary to take into account that we are technological subjects that develop economically on the transformation of nature".

»> The authors miss the chance to reference and summarise other literatures that support the idea that the relevant entity that might be pushing the Earth into a new

system state is an assemblage comprising biological humans plus the exteriorisations of culture and technology. Some better examples would help too – that of fracking and Earth tremors doesn't seem to work so well, as it is not evidence of a proper systemic shift; there are many better examples in the canonical Anthropocene science papers, such as the shaping of rivers (e.g. http://dx.doi.org/10.1016/j.ancene.2015.03.003).

We think the reviewer is right, river reshaping is historically a good example and we added other clarifying examples.

New text included:

"Nevertheless, although these new technologies as fracking should be considered under very high scrutiny, some "old" technologies such as hydraulic engineering, has already proved to have the potential of drive mayor ecosystemic changes. In fact, Williams and co-workers (2014) identify "Humans as the third evolutionary stage of biosphere engineering of rivers". For the authors, the first two bio-engineering forces are oxygenic photosynthesis and the development of vascular plants with root systems. Then in third place comes human activities such as drainage, agriculture, the construction of artificial water bodies, the development of artificial water storage and flow regulation structures and some second-order effects as changes in global-scale chemical and biogeochemical modification of terrestrial water bodies (Meybeck, 2003). Sometimes even small and apparently innocuous technology can add up to produce huge effects, which is the case of human use of chlorofluorocarbons (CFCs) often used in aerosol cans and cooling devices such as fridges, that was demonstrated were the driver of Ozone layer depletion. Discovered using 20 years of ozone levels measurements over the Antarctic stations of Halley and Faraday by Joe Farman, Brian Gardiner and Jonathan Shanklin, it was published in a foundational paper of 1985 that transformed the fields of atmospheric science and chemical kinetics, and led to global changes in environmental policy (Farman, J. C., Gardner, B. G. & Shanklin, 1985; Solomon, 2019). Even "green" technologies could lead to important planetary changes if implemented massively (Kleidon, 2016) as could happen with Eolic energy production

that at the end of the day extract kinetic energy out of climatic systems, "Large-scale exploitation of wind energy will inevitably leave an imprint in the atmosphere" (Buchanan, 2011)"

»> . . . Secondly their normative conclusion – that we therefore need to reject technology, revert to classical Homo Sapiens, and thereby stay in the Holocene – does not follow. . .

We are not saying that at all. What we are saying is that we need to take technological coupling into account when trying to understand Earth System Dynamics and that some types and intensities of technological coupling should be treated with the maximum application of the non-naive (this is key for not falling into misunderstanding) Precautionary Principle.

New text included: "A word of warning here, by no means we are proposing to neglect scientific or technological progress, nor we are thinking we should live as hunter-gatherers. We are merely saying that we need to take technological coupling into account when trying to understand Earth System Dynamics and that some types and intensities of technological coupling should be treated with the maximum application of the non-naive (this is key for not falling into misunderstanding) Precautionary Principle. As pointed out by Taleb and co-workers (2014), "a non-naive view of the precautionary principle is one in which it is only invoked when necessary, and only to prevent a certain variety of very precisely defined risks based on distinctive probabilistic structures. But, also, in such a view, the PP should never be omitted when needed". For example, in small quantities even controversial technologies as nuclear plants which we know may be prone to catastrophic accidents (Perrow, 1984) don't require to invoke PP. What Perrow notice after his analysis of the Three Mile Island nuclear accident in 1979 is that normal or systemic accidents, often catastrophic, are mainly inevitable in extremely complex systems as nuclear power plants. Nevertheless, even when terrible, the effects of one nuclear plant accident, don't propagate to other nuclear plants and most of the worst damage is local. In this sense, although it is known that the

potential harm due to not only accidents as radiation release or core meltdowns but also by radioactive waste can be large. At the same time, the nature of these risks has been extensively studied, and the risks from local uses of nuclear energy have a scale that is much smaller than global (Taleb, 2014). On the other hand we have geoengineering, an unproven new technology whose potential effects are clearly of a planetary scale and for which we don't have any understanding of direct or indirect risks. Then, to make it very clear, our approach for the use of the Technocene term is not to limit, reject or demonize technology per se, but to promote awareness to only some type of technologies depending on their use, type of risk, scale and coupling with other Earth Systems compartments. Very similar to the idea of incorporating defaunation concept and not only use the established loss of biodiversity. In addition, and maybe more important perspective is that thinking of Technocene rather than Anthropocene, also opens debate and analysis of philosophical (ontological, ethical), political and social problems about Climate Change and other components of Planetary Crisis, enhancing a deeper integral understanding of it."

»> The Shakespeare quote, 'What's in a name? That which we call a rose by any other name would smell as sweet' (line 71) seems very ill-chosen by the way – the quote is generally taken to mean that names are not important.

We thank for the heads up on this, we were not aware of that and it certainly is not a necessary quote for the argumentation of the text but only cosmetic, so we removed it.

»> On more minor matters, the English language needs work. Most sentences have at least something wrong, but the worst slips are 'Inhere' instead of 'in here' (line 17 – though in fact it would be better to delete the word altogether), 'fall out' instead of 'follow' (line 58), 'scare' instead of 'scar' (line 94).

Thank you, we corrected these and gave the manuscript a new language revision.

---

## Referee Comment (RC2) · Carsten Herrmann-Pillath (Referee) · 8 Feb 2020

The authors present ideas that have been articulated und debated in the literature for long: a) in the context of the anthropocene discussion, especially with reference to the technosphere notion (e.g. Peter Haff), even using the same term 'technocene' (for an overview, see Malhi, Yadvinder. 2017. The Concept of the Anthropocene. Annual Review of Environment and Resources 42, 1, 77–104. b) in the rich and complex literature following Dawkin's 'Extended Phenotype' book (1982), including very substantial contributions such as Kim Sterelnyi's. In that context, one should be aware that the notion

of 'culture', as used in evlutionary anthropology, includes artefacts, hence technology. Therefore, there is also a well developed modelling literature following the trail of Boyd and Richerson. This reviewer does not pay respect to modesty in pointing out that I have developed a very similar model and diagram as the authors sketch, in the book 'Foundations of Economic Evoluition' (2013), explicitly building on niche construction theory and co-evolutionary theory. That means, I do not think that the authors present a new idea. If they really want to contribute a new idea in this field, I recommend that they should concentrate on the question whether certain universal evolutionary principles apply across all ontological levels, such as thermodynamic imperatives. But even here they must build on what we already have, such as Peter Haff's contributions. There is also a rich literature on evolutionary modelling of technology which employs generic evolutionary concepts, such as the replicator notion, which the authors may find inspiring.

---

## Author Comment (AC3) · 11 Feb 2020

We thank the Carsten Herrmann-Pillath for his comments that we respond below.

Comments (»>)

»> The authors present ideas that have been articulated und debated in the literature for long: a) in the context of the anthropocene discussion, especially with reference to the technosphere notion (e.g. Peter Haff), even using the same term 'technocene' (for an overview, see Malhi, Yadvinder. 2017. The Concept of the Anthropocene. Annual

[Figure]

Review of Environment and Resources 42, 1, 77–104. b) in the rich and complex literature following Dawkin's 'Extended Phenotype' book (1982), including very substantial contributions such as Kim Sterelnyi's. In that context, one should be aware that the notion of 'culture', as used in evlutionary anthropology, includes artefacts, hence technology. Therefore, there is also a well developed modelling literature following the trail of Boyd and Richerson.

We thank the Reviewer for sharing this literature we have not included basically because this is not a systematic review of the literature, our focus is on Systems Dynamics, and the format of the article type is very length restricted, but we will make our best to reflect some of this.

New text added:In this sense, the discussion in the literature about the concept of anthropocene is important. For example, from the ecosystem sciences Yadvinder Malhi (2017) explores the functioning of the biosphere and its interactions with global change; while from a cultural evolution perspective Boyd, R., & Richerson, P. J. (1996) have studied the development of this geological era. This, without neglecting Haff's vision, Peter K. (2014) who also proposes six key rules that mediate human beings and technology in the anthropocene.

»>This reviewer does not pay respect to modesty in pointing out that I have developed a very similar model and diagram as the authors sketch, in the book 'Foundations of Economic Evoluition' (2013), explicitly building on niche construction theory and co-evolutionary theory. That means, I do not think that the authors present a new idea. If they really want to contribute a new idea in this field,

New text added: Of particular interest is the work of Peter Halff about how different human technological systems such as communication, transportation, bureaucratic and other systems are interlinked and actually act to metabolize energy (mostly fossil fuels) in a sort or global emergent entity with similarities to the lithosphere, atmosphere, hydrosphere and biosphere. The author calls this the Technosphere, which he considers the defining system of the Anthropocene and most important in the context of the present work, he thinks it influentiate and even model what we might consider most intimately and essentially human: our ideas, personal purposes, feelings, and dreams. In the same sense a direct antecedent of the importance of niche construction is the seminal work of Herrmann-Pillath who has pointed out how technology co-evolve with other components of human culture such as its institutions in parallel with behavioral and biological evolution, constituting a key element of niche construction. This recognition is incorporated in what we think is a novel ontology, the Ecobiont, discussed in our previous work (López-Corona, et.al. 2019) and that makes our understanding of Technocene somehow different to previous proposals, because is not only about the predominium of technology that enhance human capacity for niche construction; it is also about the predominium of a new ecobion, the technobiont.

»> I recommend that they should concentrate on the question whether certain universal evolutionary principles apply across all ontological levels, such as thermodynamic imperatives. But even here they must build on what we already have, such as Peter Haff's contributions. There is also a rich literature on evolutionary modelling of technology which employs generic evolutionary concepts, such as the replicator notion, which the authors may find inspiring.

Thank you for the recommendation, but we think that is an entirely different work. As we see it, our contribution is well pose under: (1) Thechnocne as a preponderance of new ecobiont; (2) Technocene concepts implies the use of Precautionary Principle in relation to the space of possible interventions for Planetary Crisis fight; (3) turning attention to a topic that has not received enough consideration, the great technological acceleration of the past 50 years and how it has become an Earth System Dynamics changer